# Neuroprotective and Anti-Inflammatory Activities of Hybrid Small-Molecule SA-10 in Ischemia/Reperfusion-Induced Retinal Neuronal Injury Models

**DOI:** 10.3390/cells13050396

**Published:** 2024-02-25

**Authors:** Charles E. Amankwa, Lorea Gamboa Acha, Adnan Dibas, Sai H. Chavala, Steven Roth, Biji Mathew, Suchismita Acharya

**Affiliations:** 1North Texas Eye Research Institute, University of North Texas Health Science Center, Fort Worth, TX 76107, USA; charlesamankwa@my.unthsc.edu (C.E.A.); dibasa@yahoo.com (A.D.); schavala@gmail.com (S.H.C.); 2Department of Pharmacology and Neuroscience, University of North Texas Health Science Center, Fort Worth, TX 76107, USA; 3Department of Anesthesiology, College of Medicine, University of Illinois at Chicago, Chicago, IL 60612, USA; loreagamboa1996@gmail.com (L.G.A.); rothgas@uic.edu (S.R.)

**Keywords:** oxidative stress, SA-10, NO donor, hybrid antioxidants, inflammation, microglia, ischemia reperfusion, retinopathy, neuronal cell death

## Abstract

Embolism, hyperglycemia, high intraocular pressure-induced increased reactive oxygen species (ROS) production, and microglial activation result in endothelial/retinal ganglion cell death. Here, we conducted in vitro and in vivo ischemia/reperfusion (I/R) efficacy studies of a hybrid antioxidant–nitric oxide donor small molecule, **SA-10,** to assess its therapeutic potential for ocular stroke. Methods: To induce I/R injury and inflammation, we subjected R28 and primary microglial cells to oxygen glucose deprivation (OGD) for 6 h in vitro or treated these cells with a cocktail of TNF-α, IL-1β and IFN-γ for 1 h, followed by the addition of **SA-10** (10 µM). Inhibition of microglial activation, ROS scavenging, cytoprotective and anti-inflammatory activities were measured. In vivo I/R-injured mouse retinas were treated with either PBS or **SA-10** (2%) intravitreally, and pattern electroretinogram (ERG), spectral-domain optical coherence tomography, flash ERG and retinal immunocytochemistry were performed. Results: **SA-10** significantly inhibited microglial activation and inflammation in vitro. Compared to the control, the compound **SA-10** significantly attenuated cell death in both microglia (43% vs. 13%) and R28 cells (52% vs. 17%), decreased ROS (38% vs. 68%) production in retinal microglia cells, preserved neural retinal function and increased SOD1 in mouse eyes. Conclusion: **SA-10** is protective to retinal neurons by decreasing oxidative stress and inflammatory cytokines.

## 1. Introduction

Ischemia/reperfusion injury is a common cause of irreversible blindness globally. It is a hallmark pathological feature in many retinal neurodegenerative diseases, including diabetic retinopathy (DR), dry age-related macular degeneration (AMD), glaucoma and central retinal artery occlusion (CRAO) [1,2,3]. CRAO, an analogue to cerebral stroke in the eye, is an ocular emergency that causes visual morbidity and is often linked to embolism in older people [4]. The transient occlusion of the central retinal artery obstructs blood supply to the retina, resulting in acute oxygen deprivation, limitation of metabolic substrates and energy depletion. Consequently, this results in malfunction of the Na^+^/K^+^ ATPase pump, neuronal depolarization and, ultimately, apoptosis of retinal ganglion cells (RGCs) [5]. Since the retina’s high metabolic demand is met by an abundant supply of nutrients and oxygen from the retinal vasculature, ischemia-induced disruption of blood supply to the retina results in profound morphological, functional and vascular changes, both in the retina and the optic nerve head region. Interestingly, the precise pathophysiology of retinal ischemia/reperfusion injury remains incompletely understood. However, oxidative stress and neuroinflammation have been reported as key players in its progression [6]. Ischemia–reperfusion events create an imbalanced hypoxic/hyperoxic environment, resulting in the retina increase in reactive oxygen species (ROS) and sequestration of immune cells/proinflammatory mediators, like tumor necrosis factor-alpha (TNF-α), interleukin-1 beta (IL-1β) and interferon gamma (IFN-γ), thereby initiating the inflammatory cascade and leading to neuronal cell death. Moreover, the imbalance between the rate of oxygen free radicals generated and natural cellular antioxidants in the ischemia/reperfusion state leads to detrimental interactions with cellular macromolecules, leading to lipid peroxidation, protein modification and nucleic acid breakdown [7].

Currently, there is no cure for ischemia/reperfusion-induced retinal injury other than the standard therapeutic approach of restoring blood flow to salvage hypo-perfused tissues. However, efficient reperfusion of tissues is not guaranteed, and the limited tolerance of neurons to hypoxic stress imposes a restricted time window for effective reperfusion therapy [7,8]. Moreover, studies examining the consequences of inflammatory gene expression and therapeutic alternatives to retinal IR injury are limited. Hence, there remains an unmet need to identify potential therapeutic agents that can address the intricate pathophysiology of ischemia/reperfusion retinal injury. Notably, antioxidants, including vitamins C and E, have shown promising therapeutic benefits in retinal ischemic injury and rodent stroke models [9,10,11]. Additionally, free radical scavengers, like NXY-059 (disodium 2,4-sulphophenyl-*N*-tert-butylnitrone, Cerovive) and thiol-containing *N*-acetyl cysteine, have proven effective in reducing the infarct size and improving neurological outcomes in brain stroke models [12,13].

In our previous studies, the first-generation multifunctional hybrid small-molecule **SA-2** with nitric oxide (NO)-donating and antioxidant properties demonstrated significant neuroprotection in three different acute RGC injury models [14]. Building on this, the second-generation hybrid NO donor–antioxidant molecule **SA-9** and its active sulfoxide metabolite **SA-10** were synthesized, which exhibited broad-spectrum ROS-scavenging activity and efficient cytoprotection against oxidative stress in trabecular meshwork (TM) cells. Notably, the sulfone compound **SA-10** was found to be a potent broad-spectrum (superoxides (O_2_^−^) and hypoxyl radicals (HOCl)) ROS scavenger and NO donor [15]. In addition, **SA-10** and a nano-encapsulated formulation of **SA-10** showed significant protective effects on endothelial cells, and decreased inflammation and increased blood perfusion in a mouse hind limb ischemia model [16].

In the current study, we explored the anti-inflammatory and neuroprotective efficacy of **SA-10** using an established in vitro model of oxygen glucose deprivation/reperfusion (OGD/R) with retinal cells and a microglial activation model. We further assessed the in vivo neuroprotective effects of intravitreally injected **SA-10** in a mouse model of retinal ischemia/reperfusion injury.

## 2. Materials and Methods

### 2.1. Chemicals and Reagents

The compounds SIN-1 (a known NO donor) and **SA-10** were synthesized following an in-house developed protocol that was previously published [15]. The reagents included Hank’s Balanced Salt Solution (HBSS, #14025092, ThermoFisher, Gibco, Waltham, MA, USA); 1% ovomucoid (#T9253, Sigma-Aldrich, St. Louis, MO, USA); Dulbecco’s phosphate-buffered saline (DPBS) (#194146, ThermoFisher, Gibco, Waltham, MA, USA); a rapidly growing mycobacterial (RGM) medium containing 250 μg/mL of epidermal growth factor (EGF, #10770-910, Prep-Tech, Kingston, PA, USA), 100 μg/mL of fibroblast growth factor (FGF, #,10018B, Prep-Tech, Kingston, PA, USA), 50 μg/mL of brain-derived neurotrophic factor (BDNF, #450-02, Prep-Tech), and 100 μg/mL of neurotrophic factor-3 (NT-3, #450-03, Prep-Tech); Dulbecco’s modified Eagle’s medium (DMEM): F12 (1:1) medium (#10565, GIBCO); 10% FBS (F4135, Millipore-Sigma, St. Louis, WA, USA) /1% penicillin/streptomycin (#15640, Gibco); a multiplex rat cytokine chemokine assay kit (Millipore Sigma); lactate dehydrogenase (LDH) (Promega, Madison, WI, USA); 3-(4,5-dimethylthiazol-2-yl)-5-(3-carboxymethoxyphenyl)-2-(4-sulfophenyl)-2H-tetrazolium (MTS) (Promega, Madison, WI, USA); dichlorodihydrofluorescein diacetate (DCFDA); TRIzol (Invitrogen, Carlsbad, CA, USA); Direct-Zol RNA microPrep (Zymo Research, Irvine, CA, USA); a cDNA Reverse Transcription Kit (Applied Biosystems, Foster City, CA, USA); 0.1% Tropicamide (Akorn, Inc., Lake Forest, IL, USA); mouse anti-βIII-Tubulin antibody (Sigma-Aldrich, St. Louis, MO, USA and diluted 1:1000); rabbit polyclonal to superoxide dismutase 1 (SOD1) (ab13498, diluted 1:1000, Cambridge, UK); and mouse monoclonal anti-3-Nitrotyrosine antibody (ab110282, 1 µg/mL, Cambridge, UK).

### 2.2. Retinal R28 Cell Line

The R28 retinal cell line was purchased from Kerafast (Boston, MA, USA) and cultured according to the supplier’s instructions, as we have previously reported [17]. R28 is an adherent retinal precursor cell line derived from postnatal day 6 Sprague Dawley rat retina [18]. Retinal R28 cells express all neuronal and astrocyte markers (β tubulin III, calbindin, syntaxin, neurofilament, PKC, GFAP and vimentin) [19].

### 2.3. Primary Rat Retinal Neural Cell Isolation and Culture

Primary neurons were isolated and cultured from neonatal E19-P1 rat pups, as previously described [20]. Briefly, rat eyeballs were enucleated, and retinal tissues were isolated and incubated in Hank’s Balanced Salt Solution (#14025092, Thermo Fisher, Gibco, Waltham, MA, USA) containing 10 U/mL of papain, 0.2 mg/mL of L-cysteine, and 0.4% DNase I for 5–8 min at 37 °C. The eyeballs were transferred to an ovomucoid solution comprising 0.0.4% DNAse and 1% ovomucoid (#T9253, Sigma-Aldrich) in DPBS (#194146, Thermo Fisher, Gibco, Waltham, MA, USA). This step was to completely quench residual papain activity. Afterwards, retinal pieces were gently triturated, forming a unicellular suspension. Following centrifugation at 200× *g* for 11 min, cells were resuspended in a rapidly growing mycobacterial (RGM) medium containing 250 μg/mL of epidermal growth factor (EGF, #10770-910, Prep-Tech), 100 μg/mL of fibroblast growth factor (FGF, #,10018B, Prep-Tech), 50 μg/mL of brain-derived neurotrophic factor (BDNF, #450-02, Prep-Tech), and 100 μg/mL of neurotrophic factor-3 (NT-3, #450-03, Prep-Tech). Retinal neural cells were then plated on poly-D-Lysine and laminin pre-coated cover slips and incubated at 37 °C with 8% CO_2_. Subsequently, cells were maintained via partial (50%) media change in every 3–4 days and utilized within 2 weeks.

### 2.4. Primary Rat Retinal Microglial Culture and Microglial Activation

Primary retinal microglial cells (MCs) were isolated from postnatal (P1-3) rat pups following a published protocol [21]. Briefly, the collected retinal tissues were mechanically dissociated by pipetting up and down and centrifuged to collect cells. Cells were re-suspended in DMEM: F12 (1:1) medium (#10565, Thermo Fisher, Gibco, Waltham, MA, USA) and 10% FBS (F4135, Millipore-Sigma, St. Louis, WA, USA) /1% penicillin/streptomycin (#15640, Thermo Fisher, Gibco, Waltham, MA, USA), and plated in poly-D-Lysine-coated T75 flasks at a density of 2 M cells/flask (#353136, Fisher Scientific, Chicago, IL, USA). To ensure proper maintenance of adequate cell growth, the MC cell medium was replaced freshly on days 1 and 7. After 14 days, MCs were detached from the astrocyte monolayer by gentle shaking and plated in 96-well plates with DMEM-F12/10% FBS. Microglial activation was induced the next day by treating cells for 6 h with a cytokine mixture (“TII”: 10 UI/mL of IFN-γ, 10 ng/mL of TNFα, and 10 ng/mL of IL-1β), as we have described earlier [22]. Afterwards, the cytokine mixture was washed and incubated with **SA-10** for 18 h. Total nitrite levels were measured as an indirect measure of nitric oxide synthase 2 expression and activity [22] Reactive oxygen species (ROS) and cytokine levels were also assessed as previously described using conditioned media [17,19].

### 2.5. In Vitro Ischemia/Reperfusion via Oxygen Glucose Deprivation/Reperfusion (OGD/R) Model

To create an in vitro model of retinal ischemia/reperfusion, we used oxygen glucose deprivation (OGD) followed by reperfusion in both R28 and primary retinal neural cells. R28 cells were plated to reach 70% confluence in a normal medium. For the OGD/R model, cells were cultured in a glucose-free medium and subjected to hypoxia (1% O_2_, 5% CO_2_) for 4 h. Cells were then re-oxygenated (21% O_2_, 5% CO_2_) for another 18 h. Cells from the OGD/R model and normoxic controls were then assayed for lactate dehydrogenase (LDH) (Promega, Madison, WI, USA) and cell proliferation (MTS, Promega, Madison, WI, USA).

### 2.6. Cytotoxicity, Cell Proliferation, Nitrite and ROS Measurement Assays

Cytotoxicity was determined by using a Sytox non-radioactive cytotoxicity assay kit (Promega, Madison, WI, USA). Briefly, cultured supernatant samples from normoxic and OGD/R cells treated with the compounds were transferred to a 96-well plate, and an equal volume of Sytox reagent was added; the samples were incubated for 30 min at room temperature, and absorbance was measured at 490 nm. Percentage cytotoxicity was determined from the % of LDH released into the supernatant. 

Cell proliferation was measured by CellTiter 96^®^ AQ_ueous_ Non-Radioactive Cell Proliferation Assay kit (Promega, Madison, WI, USA). The assay reagent (MTS/ phenazine methosulfate (PMS) at 20 µL) was added to cells growing in a 96-well plate under normoxic and OGD conditions, which were treated with the compounds and incubated at 37 °C for 1–4 h. Absorbance was measured at 490 nm, and cell proliferation is reported as the corrected absorbance at 490 nm. The nitrate in the cell culture medium was detected by Griess reagent, following the protocol provided with the detection kit. At the end of the experiment, the cell culture supernatants from the treated and control groups were mixed with an equal volume of Griess reagent, and absorbance was measured at 540 nm. Increased absorbance correlated with increased production of total nitrite NO. The amount of ROS released from microglia was measured by using a cellular DCFDA assay kit (ab113851, Abcam, Cambridge, MA, USA), following the manufacturer’s protocol. Briefly, microglial cells were plated onto 96-well plates and subjected to TII insult for 2 h, followed by treatment with either 10 µM of SIN-1 or **SA-10**. The DCFDA reagent was added at the end of the experiment, followed by incubation for 45 min, and fluorescence intensity was measured.

### 2.7. Cytokine and Chemokine Measurement Assays

Cytokines and chemokines secreted into the cell culture supernatant from microglia were assayed with a multiplex rat cytokine chemokine assay kit (Millipore Sigma, St. Louis, MO, USA), following the manufacturer’s instructions. Five cytokines, including proinflammatory and anti-inflammatory cytokines (interleukin (IL-1β, IL-4, IL-6, IL-10 and tumor necrosis factor (TNF-α), and MCP-1 were analyzed. Results were compared across groups in which equal numbers of cells were evaluated. The assays were performed in triplicate using Millipore multi-screen 96-well plates. Data were collected using MagPix (Luminex, Austin, TX, USA). Data analysis was performed using the Millipore immunoassay curve fitting software Belysa 1.1 for Luminex XMAP ™. A five-parameter logistic formula was used to calculate the sample concentrations from the standard curves.

### 2.8. qPCR Analysis of Inflammatory Mediators, IL-1β and iNOS

To check the anti-inflammatory effect of **SA-10**, we conducted q-PCR analysis of IL-β and inducible nitric oxide synthase (iNOS) gene expression in retinal neurons post treatment as described previously [17]. Total RNA was isolated using TRIzol (Invitrogen, Carlsbad, CA, USA) and Direct-Zol RNA microPrep (Zymo Research, Irvine, CA, USA) as previously published [17]. cDNAs were generated using a cDNA Reverse Transcription Kit (Applied Biosystems, Foster City, CA, USA), and subjected to PCR amplification using the primer pairs for IL-β and iNOS listed in Table 1 [17]. Relative expression levels were quantitated using the ΔΔCt method with the internal control (GAPDH) normalized to the mRNA ratio levels.

### 2.9. Animals

All animal studies were performed in accordance with the ARVO Statement for the Use of Animals in Ophthalmic and Vision Research and the protocol approved by the University of North Texas Health Science Center Institutional Animal Care and Use Committee (IACUC 2019-0036). For the ischemia/reperfusion study, we used 3-to-4-months-old C57BL/6J mice from Charles River (Wilmington, MA, USA). For primary cell isolation, we used pregnant rats from Harlan, Indianapolis, IN, USA. The procedures were conformed to the NIH Guide for Care and Use of Animals in Research and were approved by the University of Illinois at Chicago’s Animal Care and Use Committee.

### 2.10. Ischemic/Reperfusion Injury Model in Mouse Eyes

C57Bl6/J mice (12 weeks, *n* = 5) were anesthetized and treated with proparacaine prior to performing the ischemia/reperfusion (I/R) injury on the left eye, as previously described by us [14]. Afterwards, an intravitreal injection of 2 µL of 2% **SA-10** formulated in PBS was administered to I/R-induced (I/R + **SA-10**) mouse eyes, and 2 µL of PBS alone was administered to another group of I/R-induced (I/R + PBS group) eyes on day 0. On day 7, pattern electroretinography (ERG) was performed with the JORVEC System (Jorvec Inc., Miami, FL, USA) to measure RGC function using the right eye as the contralateral control. Spectral-domain optical coherence tomography (SD-OCT) was conducted to measure the change in the overall thickness of retina layers, and flash ERG was performed to measure the function of bipolar cells (b-wave) as well as photoreceptor cells (a-wave) after 28 days.

### 2.11. Pattern Electroretinography (PERG)

PERG was utilized to assess RGC function by measuring PERG waveform amplitude following I/R injury in mice, as described previously [23]. The mice were anesthetized and placed on a heated stage. PERG responses were evoked in response to contrast reversal of patterned visual stimuli using a commercially available PERG system (Jorvec Inc., Miami FL, USA). In the red-light environment, the PERG responses were acquired via a needle electrode placed sub-dermally in the mouse snout, while the reference electrode was placed at the base of the head, and the ground electrode at the base of the tail. Each animal was positioned at 11 cm from a monitor displaying full-field pattern stimuli (45° radius visual angle subtended on full-field pattern, 2 reversals per second, 300 averaged signals with cut-off filter frequencies of 1–30 Hz, 98% contrast, and 800 cd/m^2^ average monitor illumination intensity) without dark adaptation. This was to exclude the possible effect of direct photoreceptor-derived evoked responses. PERG amplitudes at baseline and 7 days post I/R injury were calculated according to a previously described method [24].

### 2.12. Spectral-Domain Optical Coherence Tomography (SD-OCT)

SD-OCT was conducted as previously published by us [14] using the Reveal OCT2 Imaging System (Phoenix Research Labs, Pleasanton, CA, USA) with a contact lens specifically designed for mice. The mice were anesthetized with isoflurane inhalation and pupillary dilations were achieved with the instillation of a drop of 0.1% Tropicamide (Akorn, Inc., Lake Forest, IL, USA). The corneas were lubricated with GenTeal liquid gel (Novartis, East Hanover, NJ, USA) to prevent drying, and ocular fundus imaging was conducted with a fundus camera of the Micron IV imaging system (Phoenix Research Laboratories, Pleasanton, CA, USA). Three positions of retinal OCT images from the same eye were set horizontally across the optic disc, with one disc diameter superior and one inferior to the optic disc. To eliminate artifacts, ten to twenty images were averaged at each location.

### 2.13. Electroretinogram (ERG)

The animals were dark adapted and kept under dim red light prior to ERG measurements. The mice were anesthetized by isoflurane inhalation after 12 h of dark adaptation. Using 0.1% Tropicamide eye drops, their pupils were dilated, and the corneal surface was kept moist with GenTeal liquid gel (Alcon Laboratories Incorporated, Fort Worth, TX, USA). During the measurements, the mice were kept on a heating pad to maintain an optimal body temperature. One needle probe was inserted subcutaneously for reference between the two eyes. A silver needle placed in the proximal part of the tail served as the ground electrode. The contact lens electrode (Micron IV Ganzfield ERG, Phoenix Research Laboratories, Pleasanton, CA, USA) was placed directly over the cornea. The light stimulus intensities were set as −1, 0, 1 and 1.5 log (cd.s/m^2^) for the measurements of scotopic response. Five sweeps were recorded at each light intensity with a sufficient delay between each sweep, and the average of five responses to the same stimulus intensity was used as the final waveform for a certain light stimulus. The amplitudes of both a-wave and b-wave were analyzed.

### 2.14. Immunohistochemistry

Immunohistochemistry was performed on mouse retinal sections as described previously [14]. Briefly, 5 μm thick mouse sagittal retinal sections through the optic nerve head were obtained from the mouse eyes 28 days post I/R + PBS (*n* = 3) and I/R + **SA-10** (*n* = 3) treatments. The sections were deparaffinized in xylene (Fisher Scientific) and re-hydrated using a series of ethanol washes (100, 95, 90, 80 and 50%). Using 5% normal donkey serum and 5% BSA in PBS, blocking was performed to prevent non-specific binding of the secondary antibodies. The sections were incubated with the primary antibodies, including mouse anti-βIII-Tubulin antibody (Sigma-Aldrich, St. Louis, MO, USA, and diluted 1:1000), rabbit polyclonal to superoxide dismutase 1 (SOD1) (ab13498, diluted 1:1000) or mouse monoclonal anti-3-Nitrotyrosine antibody (ab110282, 1 µg/mL), for 24 h at 4 °C [25] following blocking. With a 1: 1000 dilution of the appropriate secondary antibodies, secondary antibody incubation was carried out for 1 h. The sections in which the primary antibody incubation was omitted were used to assess non-specific staining by the secondary antibodies. Using a Zeiss LSM 510 META confocal microscope (40×) or Cytation5 (Gen5 3.12, BioTek Instruments, Winooski, VT, USA) at 20× magnification, fluorescence images were captured. A masked observer quantified the SOD1 and nitrotyrosine fluorescence intensities across the whole retinal sections using the ImageJ 1.54f software (NIH), and all results are presented as mean ± SEM.

## 3. Results

### 3.1. **SA-10** Attenuated Oxygen Glucose Deprivation/Reperfusion (OGD/R)-Induced Cell Death and Enhanced Cell Proliferation in Rat Retinal Neurons

To examine the activity of **SA-10** (Figure 1) in the OGD/R model, R28 cells were subjected to oxygen glucose deprivation for 4 h to mimic a hypoxic or ischemic environment, and subsequently reperfused with oxygen and glucose for 18 h to induce I/R injury in vitro. Using both R28 and primary rat neuronal cells, we noticed a marked protection of **SA-10** from OGD/R-induced cell death. In R28 rat mixed neuronal cells, we noticed approximately 70–80% cell death in the control cells after the OGD/R insult, as measured by LDH release (Figure 2A), which was consistent with our previously published data [19,26]. The addition of 10 µM of the compounds SIN-1 (Figure 1) and **SA-10** significantly reduced cell death. Additionally, we evaluated cell proliferation using an MTS cell proliferation assay by subjecting R28 cells to OGD/R due to its inherent cell proliferative effect. The results presented in Figure 2B demonstrate that **SA-10** significantly increased cell proliferation compared to OGD/R (labelled as OGD) alone and OGD/R + SIN-1. Remarkably increased proliferation was observed with **SA-10** compared to SIN-1 (a known NO donor), demonstrating the enhanced protective effect due to the hybrid property. We repeated these experiments using cultured primary mixed retinal neuronal cells isolated from juvenile rat pup eyes. Altogether, as shown in Figure 2C,D, the hybrid compound **SA-10** with both NO-donating and ROS-scavenging abilities was more potent in decreasing cell death (Figure 2C) and improving cell proliferation (Figure 2D), whereas only the NO donor SIN-1 was not effective in improving the proliferation of neural cells, thereby indicating the superior neuroprotective activity of **SA-10** compared to SIN-1.

### 3.2. **SA-10** Decreased OGD/R-Induced Inflammation in Primary Rat Retinal Neural Cells

To evaluate the inflammatory activation of retinal neurons, we checked the mRNA expression levels of inducible nitric oxide (iNOS) and the proinflammatory cytokine IL-1β in primary mixed retinal neurons subjected to OGD/R. Treatment with **SA-10** (10 µM) significantly decreased the relative % change in mRNA expression for iNOS and IL-1β compared to OGD cells (Figure 3A,B), demonstrating significant anti-inflammatory activity.

### 3.3. **SA-10** Attenuated Pathologically Activated Microglial Cell Death, Total Nitrite Levels and ROS Production

Pathologically activated retinal microglial cells in an ischemic eye release reactive oxygen and nitrogen species and increase proinflammatory mediators. The primary rat retinal microglial cells activated by TII (a cocktail of inflammatory mediators, including TNF-α, IL-1β and IFN-γ) resulted in increased microglial cell death, with elevated ROS and nitrite levels (Figure 4). Here, we observed that treatment with SIN-1 or the hybrid compound **SA-10** significantly decreased the total nitrite production and microglial death (Figure 4A,B). The addition of SIN-1 (an NO donor) did not reduce the ROS level; however, **SA-10** significantly decreased the levels of ROS when compared to the control (Figure 4C). These data demonstrate the remarkable antioxidant-induced cytoprotective effect of the NO donor/antioxidant hybrid compound **SA-10** by attenuating the activation of microglia.

### 3.4. **SA-10** Decreased Proinflammatory Molecules and Increased Anti-Inflammatory Cytokines in Activated Microglia

Using multiplex ELISA, we determined the changes in proinflammatory and anti-inflammatory molecules from the supernatant of activated retinal microglia with or without treatment of 10 µM of either SIN-1 or **SA-10**. As shown in Figure 5A, while TII significantly increased the proinflammatory cytokines IL-1β, TNF-α and IL-6 and macrophage chemoattractant protein-1 (MCP-1), treatment with **SA-10** significantly decreased all proinflammatory cytokines. IL-10, an anti-inflammatory cytokine, decreased in TII-activated microglia, and **SA-10** significantly increased the IL-10 level (Figure 5B). On the other hand, the NO donor compound SIN-1 was not effective in reducing TII-induced IL-1β, IL-6 or MCP-1 levels. These findings indicate that **SA-10** exhibits anti-inflammatory properties in the activation of microglia, making it a promising candidate for addressing inflammation-induced retinal conditions.

### 3.5. Intravitreal (ivt) Injection of **SA-10** was Protective to Retina after Ischemia/Reperfusion (I/R) Retinal Injury in Mouse Eyes

The function of retinal ganglion cells (RGCs) can be non-invasively assessed by the pattern electroretinogram (PERG) [27] technique, which measures the activity of inner retinal neurons. The amplitude of PERG reflects the degree of apparent damage to RGCs. We performed an in vivo validation study in mice using an I/R model of ocular stroke. Following a single intravitreal injection of 2% **SA-10** (78 µM) on the left eye that had undergone the I/R procedure, standard PERG was analyzed 7 days post I/R injury to determine whether the compound **SA-10** could prevent PERG deficits (Figure 6A,B). Spectral-domain optical coherence tomography (SD-OCT) was performed to measure the change in the overall thickness of retina layers (Figure 6C), and flash ERG was performed to measure the function of other retinal neural cells, including bipolar cells (b-wave) as well as photoreceptor cells (a-wave), at 28 days post I/R. The right eye served as the contralateral control. The dose selection was based on our previous dosing experience with our first-generation hybrid compound **SA-2**, which has a similar physicochemical profile to **SA-10**. The compound **SA-2** (2%, 75 µM) was bioavailable in 0.3-0.9 nM concentration in the retina and choroid + scleral tissue after a single ivt injection and was effective in protecting RGCs from cell death in an optic nerve crush mouse model [14]. Therefore, we used 78 µM (2%) of **SA-10** in our experiment.

In the current I/R experiment, the sham control mice were not exposed to cannulation of the anterior chamber and did not display any evidence of functional RGC deficits at 1 week (26.7 ± 0.7 µV, *n* = 5) after surgery. However, there was a significant decrease in PERG amplitudes in the PBS-treated mice compared to the sham mice (14.8 ± 1.4 µV, *n* = 5). Intravitreal injection of a single dose of **SA-10** (2%, 78 µM) increased PERG amplitudes compared to the PBS-treated I/R mice (21 ± 1.4 µV, *n* = 5, *p* < 0.01). Retinal ischemia/reperfusion in the PBS-treated group also resulted in a significant reduction in the amplitudes of both a- and b-waves of the electroretinogram (ERG) compared to the sham control mice, as shown in Figure 6C. The compound **SA-10** significantly reversed these damages, which was further confirmed by the improved whole retinal layer thickness as measured by SD-OCT (Figure 6D).

### 3.6. Effect of **SA-10** on Superoxide Dismutase 1 (SOD1) and Protein Nitro-Tyrosylation in Mouse Retina after I/R Injury

To further evaluate the change in the levels of SOD1 enzyme in the retina, mouse retinal sections were subjected to an immunohistochemical analysis. As seen in Figure 7, significantly increased SOD-1 immunostaining (second panel, in green) was detected in the I/R + **SA-10**-treated retinas, mainly in the ganglion cell layer, the inner plexiform layer and the outer plexiform layer, in comparison with the I/R and PBS-treated group, in which minimal SOD1 staining was observed.

Since the compound **SA-10** is a nitric oxide-releasing prodrug, we performed immunolabeling of the retinas with an anti-nitrotyrosine antibody to rule out the possibility of unwanted protein nitrotyrosylation in the retinas after **SA-10** administration. As expected, we did not observe any significant increase in nitrotyrosine level in the I/R + **SA-10**-treated eyes compared to the I/R-injured and PBS-treated eyes (Figure 7, first panel in red and bar graph).

## 4. Discussion

The multifaceted pathology associated with ischemic or hemorrhagic stroke includes decreased blood circulation, thrombosis and, ultimately, cell death. The current treatment regimen is mostly directed towards improving blood circulation using vasodilators and anti-thrombotic agents; however, these interventions do not prevent/treat ischemic cell death effectively. Hence, there remains an unmet need for discovering new agents that will not only increase blood perfusion, but also prevent endothelial dysfunction, provide neuroprotection, and exert safe immunomodulatory effects for the management of ischemic stroke and retinal ischemia. Nitric oxide (NO) is directly implicated in vasodilation via increasing cyclic guanosine monophosphate (c-GMP) in endothelial cells. It is synthesized by three isoforms of NO synthase (NOS), namely endothelial, neuronal and inducible NOS. Endothelial nitric oxide synthetase (eNOS) plays a key role in the protection of the neurovascular system, and the activation of eNOS is required for neuroprotection against ischemic stroke in patients with diabetes [28]. The production of NO derived from eNOS around the nerve vessels is capable of regulating the tension between the cerebral vessels and plays a positive role in improving the blood supply to the brain tissue [29]. On the other hand, inducible nitric oxide synthetase (iNOS) is negatively implicated in the pathogenesis of I/R injury and cerebral ischemic injury [30]. Semi-quantitative reverse transcription polymerase chain reaction showed a marked iNOS mRNA expression in rat retinas following transient ischemia.

In this study, we clearly demonstrated that a prototypic nitric oxide donor and antioxidant hybrid molecule, **SA-10,** preserved the viability of both primary and immortalized R28 mixed rat neural cells following an OGD/R-induced insult. **SA-10** demonstrated a >2-fold neural survival compared to the known NO donor SIN-1. In addition, treatment with **SA-10** significantly decreased the levels of inflammatory iNOS mRNA expression in primary rat neural cells, with a corresponding decline in the proinflammatory cytokine IL-1β. Interestingly, these findings corroborate the studies by Szabo et al., who previously showed that mRNA for IL-1β induces iNOS expression in rat retinas following transient ischemia [31]. In addition, in situ hybridization experiments showed similarities in cellular localization of iNOS mRNA to that of IL-1β mRNA, suggesting that IL-1β initiates and promotes inflammatory immunologic events, such as the activation of glial cells and the activation of neutrophils, and as a result, it induces iNOS mRNA in these cells in an autocrine and or paracrine manner [32].

Moreover, ischemic injuries have been reported to stimulate not only IL-1β, but also the production of proinflammatory cytokines, such as tumor necrosis factor (TNF-α), monocyte chemoattractant protein-1 (MCP-1) and interferon gamma (IFN-γ), in the retina. Particularly, TNF-α and IL-1β are the main causes of neuronal and oligodendrocyte damage in the microglia of hypoxic neonatal rats [33]. To recapitulate such neuronal damage, primary rat retinal microglial cells were treated with a cocktail of proinflammatory cytokines (TII) and then treated with **SA-10** or SIN-1. The compound **SA-10** demonstrated more potent anti-inflammatory activity than SIN-1 by decreasing the concentrations of IL-1β, TNF-α, IL-6 and MCP-1, while SIN-1 increased the inflammatory marker IL-6.

Artery or vein occlusion, as seen in CRAO or in diabetic retinopathy, leads to oxidative stress in both retinal endothelial cells and retinal neural cells (RGCs) through the decreased activities of several antioxidant enzymes, including superoxide dismutase (SOD), glutathione peroxidase, catalase, hemeoxygenase-1 and thioredoxins (Trx1 and Trx2), and has been implicated in promoting RGC death [34,35,36]. Ischemic stress followed by reperfusion has also been reported to result in a burst of superoxide free radicals that induces oxidative stress in RGCs, resulting in the apoptosis of cells [37] and dysfunction of retinal endothelial cells, thus leading to poor blood circulation. Our findings, however, showed that the compound **SA-10** significantly scavenged ROS generated via TII inflammatory cocktail administration and further promoted microglial cell proliferation. It is worth noting that the generated superoxides from I/R or TII can combine with NO to form toxic peroxynitrite radical and further decrease the NO bioavailability. Hence, it is desired to neutralize/scavenge excess free radicals and maintain a physiological balance of NO under ischemic conditions to improve vasodilation and the formation new blood vessels [29,38].

Studies have shown that during the first week after an acute ischemic stroke, injection of superoxide dismutase (SOD) enzyme is protective to the retina [38]. SOD levels were significantly upregulated and localized in the cytoplasm of RGCs after the **SA-10** treatment. Immunolabeling for SOD was evident in the ganglion cell layer (GCL), inner plexiform layer (IPL) and outer plexiform layer (OPL) in the control labelled eyes and I/R + 2% **SA-10**-treated retinas, with no evidence of overt protein nitrotyrosylation. This further confirms the safety profile of **SA-10** in the retina and the improvement in the SOD enzyme activity, which augments the ROS-scavenging property of the compound **SA-10**. Moreover, the functional activity of the retina was significantly improved after treatment with **SA-10**, as demonstrated in the ERG, PERG and retinal thickness SD-OCT assessments. 

In summary, **SA-10** has previously been shown to reverse hyperinflammation and lung edema and significantly improve blood perfusion and physical endurance in acute and chronic ischemia models (*p* < 0.05) [16]. Here we demonstrated that, **SA-10** exerts antioxidant effects, immunomodulatory properties and functional preservation of the retina in the in vitro and in vivo ischemia/reperfusion retinal injury models. 

## 5. Conclusions

The compound **SA-10** was highly effective in inhibiting both OGD/R- and TII-induced cell death. Additionally, ROS and inflammatory marker production in rat microglia, and retinal neuronal cell death were inhibited by SA-10. A post-treatment of intravitreally administered **SA-10** protected the retina against ischemia/reperfusion (I/R)-induced retinal damage, as determined by the preservation of function (ERG and PERG) and retinal thickness (SD-OCT). Taken together, our results are consistent with our hypothesis that this novel hybrid compound **SA-10** is protective to retinal neurons by decreasing oxidative stress and inflammatory molecules, and possibly by improving retinal blood perfusion. Thus, **SA-10** could be a suitable preclinical therapeutic candidate in managing retinal I/R injury, ocular stroke and glaucomatous injury to the retina. Further pharmacokinetic study in larger size of eyes in either rabbits or non-human primates is warranted to evaluate the drug bioavailability and clinical translatability of **SA-10** as a future plan.

## Figures and Tables

**Figure 1 cells-13-00396-f001:**
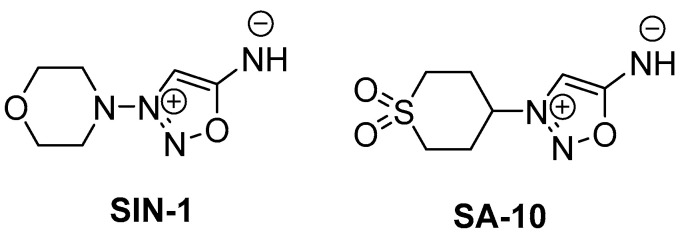
Chemical structures of **SIN-1** (a known NO donor) and **SA-10** (a hybrid NO donor and antioxidant compound).

**Figure 2 cells-13-00396-f002:**
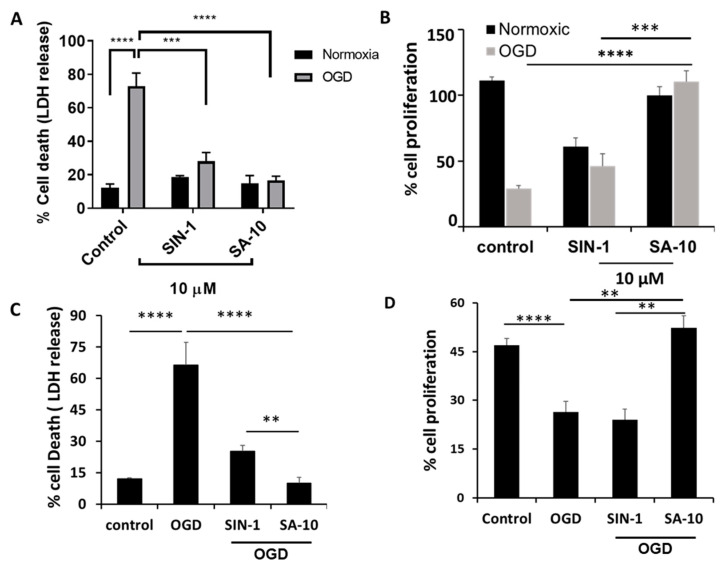
Effects of 10 µM concentration of **SA-10** and SIN-1 on R28 rat mixed neural cell and primary rat retinal mixed neuronal cell survival and proliferation after OGD/R-induced insult. (**A**,**C**) Both SIN-1 and **SA-10** at 10 µM significantly protect R28 rat mixed neuronal cells and primary rat retinal mixed neurons from OGD-induced cell death. (**B**,**D**) **SA-10** (10 µM) promotes cell survival/proliferation under OGD conditions in both cell types. Data are expressed as mean ± SD. *N* = 3, ** *p* < 0.01, *** *p* < 0.001, **** *p* < 0.0001. One-way ANOVA, GraphPad Prism v.10.1.

**Figure 3 cells-13-00396-f003:**
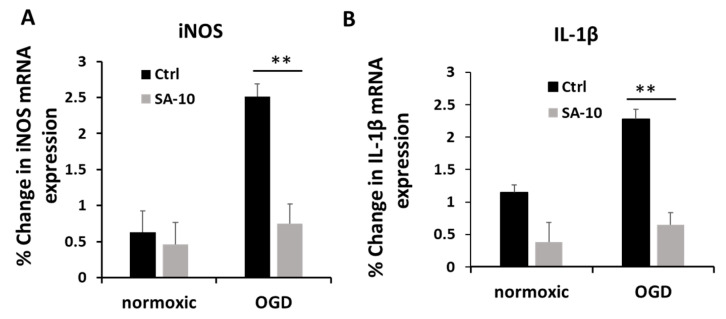
Anti-inflammatory effect of **SA-10** (10 µM) in primary rat retinal neurons. (**A**) iNOS mRNA gene expression post treatment with **SA-10** (10 µM). (**B**) Inflammatory cytokine IL-1β mRNA gene expression post treatment with **SA-10** (10 µM). Data are expressed as mean ± SD. *N* =3, ** *p* < 0.01. One-way ANOVA, GraphPad Prism v.10.1.

**Figure 4 cells-13-00396-f004:**
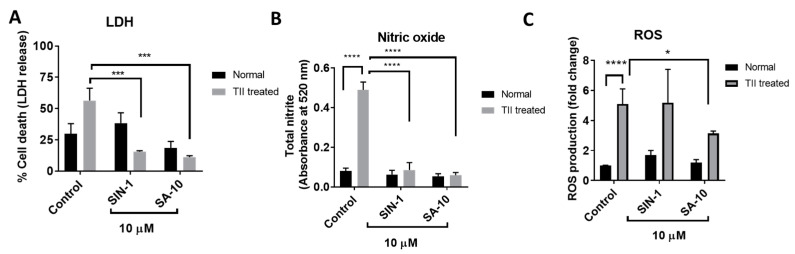
Effects of 10 µM concentration of **SA-10** and SIN-1 on cell survival, total nitrite levels and ROS in TII-induced activation of a retinal microglial model (**A**) TII-induced LDH release in rat microglial cells. Both SIN-1 and **SA-10** at 10 µM show significant protection. (**B**) Total nitrite levels are significantly increased in the TII-treated groups. **SA-10** and SIN-1 (10 µM) treatments decrease total nitrite levels in rat microglial cells. (**C**) **SA-10** (10 µM) scavenges total ROS in TII-treated and normoxic microglial cells compared to the untreated group, whereas SIN-1 does not. Data are expressed as mean ± SD. *N* = 3, * *p* < 0.05, *** *p* < 0.001, **** *p* < 0.0001. Two-way ANOVA Šídák’s multiple comparison test, GraphPad Prism v.10.1.

**Figure 5 cells-13-00396-f005:**
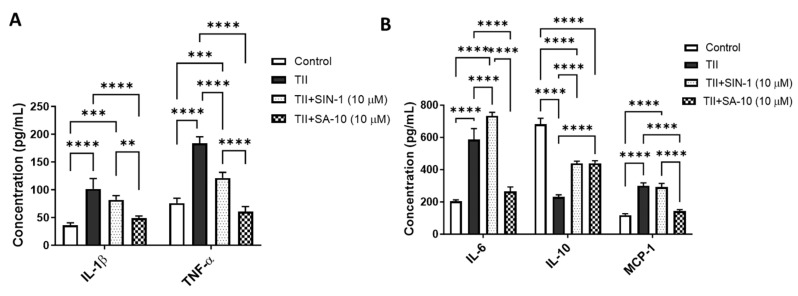
Changes in proinflammatory and anti-inflammatory molecules in activated microglia after **SA-10** treatment. (**A**) **SA-10** (10 µM) treatment significantly lowers the concentrations of proinflammatory cytokines IL-1β and TNF-α in TII-activated retinal microglial cells. (**B**) **SA-10** (10 µM) demonstrates significant decreases in IL-6 and MCP-1, whereas the anti-inflammatory cytokine IL-10 levels are relatively higher compared to the TII-treated groups. Data are expressed as mean ± SD. *N* = 3, ** *p* < 0.01, *** *p* < 0.001, **** *p* < 0.0001. Two-way ANOVA Tukey’s multiple comparison test, GraphPad Prism v.10.1.

**Figure 6 cells-13-00396-f006:**
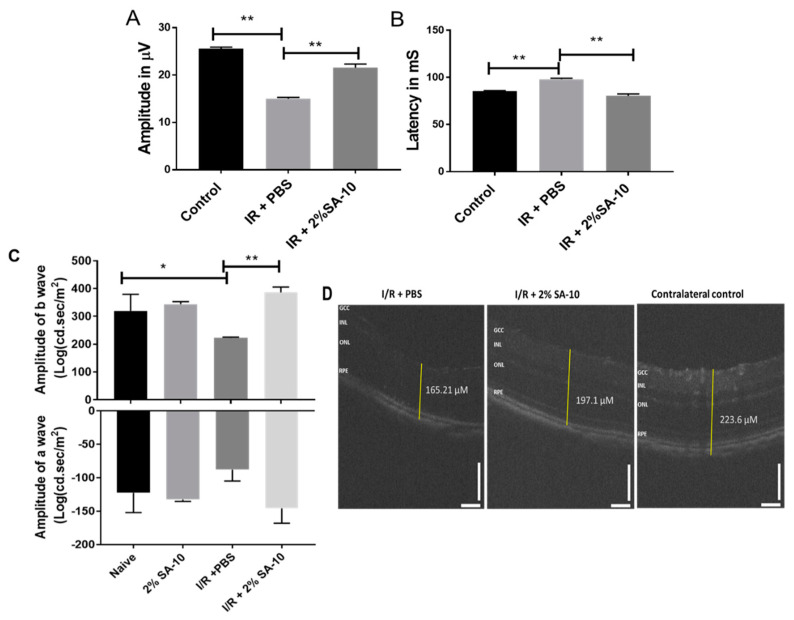
Intravitreal injection of the compound **SA-10** restores retinal function post I/R-induced retinal injury. (**A**) PERG amplitudes of eyes treated with ivt injection of PBS or with a dose of 2% **SA-10**. An analysis of the mice exposed to I/R and co-treated with **SA-10** (2%) shows significant difference in PERG amplitudes (7 days post I/R) compared to the I/R + PBS-treated mice. (**B**) Latency of PERG measured in milliseconds in mouse eyes before and after ischemia/reperfusion injury. **SA-10** increases the latency to be comparable to the untreated sham control. (**C**) Measurement of a- and b-waves by flash ERG 28 days post I/R in mice (*n* = 5). (**D**) SD-OCT of the retina showing change in total thickness of different groups. *n* = 3–6 eyes. Values are expressed as mean ± SEM, * *p* < 0.05, ** *p* < 0.01. Analysis was performed by two-way ANOVA (Tukey’s multiple comparison test) using GraphPad Prism 10.1.

**Figure 7 cells-13-00396-f007:**
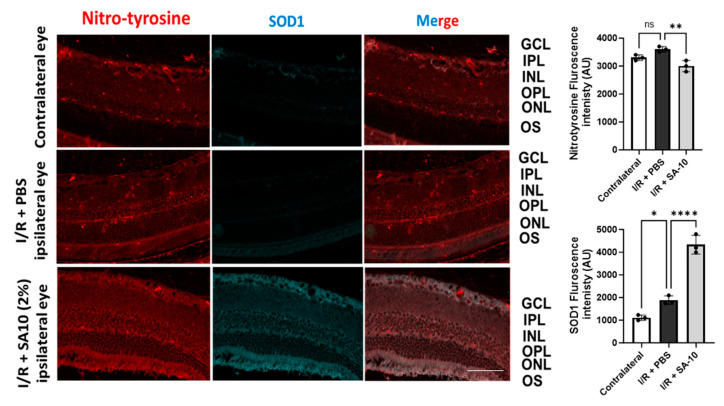
Immunohistopathology of retina. Immunohistochemical analysis for SOD1 in I/R+ PBS-treated (*n* = 3) or I/R + 2% **SA-10**-treated mouse eyes (*n* = 3). SOD1 (in green) is localized to the cytoplasm of retinal ganglion cells. Immunolabeling for SOD is significantly evident in the GCL, IPL and OPL in the control eyes and I/R + 2% **SA-10**-treated retinas. Less staining for SOD1 is observed in the I/R and PBS-treated group. Immunohistochemical analysis for nitrotyrosine in the control, I/R + PBS-treated and I/R + 2% **SA-10**-treated mouse eyes. The nitrotyrosine (in red) signal is mainly localized to the ONL, OPL, IPL, GCL and NFL in the I/R + PBS-treated or I/R + 2% **SA-10**-treated retinas. There are visible differences in the nitrotyrosine levels in the I/R + PBS-treated and I/R + 2% **SA-10**-treated retinas, and less red labeling is observed in the IR+**SA-10**-treated retinas. No significant nitrotyrosylation is observed in the control retinas, with minimal staining present in the OPL and NFL of these retinas. Values are expressed as mean ± SEM, non-significant (ns) * *p* < 0.05, ** *p* < 0.01, **** *p* < 0.0001. Analysis was performed by one-way ANOVA (Dunnett’s multiple comparison test) using GraphPad Prism v 10.1. The scale bar = 50 µm.

**Table 1 cells-13-00396-t001:** List of primers for qPCR.

Primer Pair	Sequence (5′-3′)
IL1β	GGC TGA AAT GTG GAT GGT AGA G (F) ACA AGG AAC CGT GTG GTA TTG (R)
iNOS	CAG AGA CTC CCA TTG CTT CTT (F) TTG GCC TCT CCT GTT GTA ATC (R)
GAPDH	GGG TGT GAA CCA CGA GAA AT (F) ACT GTG GTC ATG AGC CCT TC (R)

## Data Availability

The raw/processed data required to reproduce these findings will be made available upon request.

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
