# Peer review of "Neuroprotective and Anti-Inflammatory Activities of Hybrid Small-Molecule SA-10 in Ischemia/Reperfusion-Induced Retinal Neuronal Injury Models"

_cells, 2024, doi:10.3390/cells13050396_

Round 1
Reviewer 1 Report
Comments and Suggestions for Authors
Authors in this paper showed innovative hybrid compound SA-10 that exhibited protective effects on retinal neurons by reducing oxidative stress and inflammatory cytokines. the results indicated that compound SA-10 holds promising therapeutic potential for addressing retinal ischemia/reperfusion (I/R) injury associated with ischemic optic neuropathy. Good work by authors.
I have some minor issues that can be corrected to improve the quality of the manuscript.
1. The research does not address much on impact of antioxidant effect. No RNA data or Protein data (Western Blotting ) is presented to support the hypothesis. Only Sod2 immunohistochemistry data is provided.
2. The mice eye is so small and has a big lens. On day zero, intravitreal injection of 2ul (line 198) of volume of the formulations is really questionable.
3. The results also did not provide any information whether the formulations were present till the end of the experiments or how long its stability in the eye to have the effects.
Author Response
Comments and Suggestions for Authors
Authors in this paper showed innovative hybrid compound SA-10 that exhibited protective effects on retinal neurons by reducing oxidative stress and inflammatory cytokines. the results indicated that compound SA-10 holds promising therapeutic potential for addressing retinal ischemia/reperfusion (I/R) injury associated with ischemic optic neuropathy. Good work by authors.
I have some minor issues that can be corrected to improve the quality of the manuscript.
- The research does not address much on impact of antioxidant effect. No RNA data or Protein data (Western Blotting ) is presented to support the hypothesis. Only Sod2 immunohistochemistry data is provided.
Response: We thank the reviewer for your thorough review.
To clarify, we have established the antioxidant activity of SA-10 in trabecular meshwork cells in our previous publications (Amankwa C,E.,et al, Antioxidants, 2021, DOI: 10.3390/antiox10040575). There we demonstrated that, compound SA-10 exhibited superior (25%) superoxide scavenging activity in NTM-5 cell supernatants and showed a more potent and robust hypochlorouos acid radical scavenging activity. Additionally, SA-10 also decreased peroxynitrite free radical activity in-vitro. In tert-butyl hydroperoxide treated human trabecular meshwork (hTM) cells, SA-10 significantly rescued hTM cells from oxidative stress induced cell death.
Here we specifically focused on the anti-inflammatory and cyto/neuroprotective activities. Additionally, in Figure 4C we have shown significant decrease in ROS level by SA-10 in retinal microglia cells compared to activated primary microglia cells. In the mouse immunohistopathology slide (Fig. 7), we demonstrated not only there is increase in SOD1 staining in retina, but lack of nitrosylation of protein indicating decreased nitrosative stress. . Additionally, further evaluation of SA-10 in primary rat RGCs for antioxidant enzymes, and heme-oxygenase (HO-1) using western blots is currently being explored.
- The mice eye is so small and has a big lens. On day zero, intravitreal injection of 2ul (line 198) of volume of the formulations is really questionable.
Response: We agree mouse eyes are smaller than other rodents and humans. We followed a widely accepted protocol of intravitreal dosing. The vitreous volume of mouse eye is around 5-7 µL. As per NIH guideline 1-2 µL of volume should be the maximum volume injected via intravitreal injection to avoid injury and inflammation (J Vis Exp. 2021 Feb 6; (168): 10.3791/61749).
- The results also did not provide any information whether the formulations were present till the end of the experiments or how long its stability in the eye to have the effects.
Response: We understand that, the pharmacokinetic and bio-distribution are important parameters to understand the dosing and frequency of dosing. In our earlier publication, we have already published the PK profile and bio-distribution for SA-9, a structurally close and sulfide analog of SA-10, and demonstrated good bioavailability in both anterior and posterior segments after eye drop dosing. We are very confident ivt injection of SA-10 in our current study must have provided a therapeutic concentration in vitreous and retina as we see significant protection compared to the vehicle treated eyes. We have developed a method to detect SA-10 in eyes and will continue the PK study in rabbit eyes as our future plan.
Reviewer 2 Report
Comments and Suggestions for Authors
The authors conducted a extensive study and determined that the novel hybrid compound SA-10 has a protective effect on retinal neurons by reducing oxidative stress and inflammatory cytokines. I have some minor considerations (see the comments below). After these are addressed, I would recommend the manuscript for publication.
Abstract
Reduce the number of words in the abstract (420 words compared to the 200 allowed).
Line 23: In the main text, the authors did not mention that hybrid molecule SA-10 was tested in concentrations of 1 µM as stated in the abstract.
It is not clear why the authors used SA-2 in this study. Please explain.
Materials and Methods
Line 176: State the cytokines and chemokines included in the multiple assay kit.
Results
Line 279: Is the 1.5-fold increase significant? If it is not, there is no point in stating it.
Line 282: “...OGD/R.due…” remove the dot.
Line 284-285: Authors point that SA-10 siginificantly increased cell proliferation compared OGD/R+SIN-1. Mark in the Figure 2B. The same applies to the following sentence.
Line 289: It is not indicated in Figure 2D that SA-10 enhances proliferation as indicated in the section Results.
Line 291: in Figure 2D move “OGD/R” in the level of SIN-1A and SIN-10.
Line 293: Please indicate more clearly that in both tests (cytotoxicity and proliferation) 2 types of cells were used.
Lines 300-306: I think the methodology for these results is lacking. Please check.
Lines 320-321: I suggest you rather point out that SA-10 significantly reduced the level of ROS.
Line 326: “..only NO donor SIN-1 produces higher amount of NO (Fig.4B)..”
This is debatable, given that there is no difference between SIN-1 and SIN-10 and both compounds significantly reduced NO production in rat microglia cells.
Line 360: I suggest removing “ns” in Figure 5a and 5b, (it is understood if there are no asterisks that it is not significant) to make the figure more understandable.
Did the authors use a dose of 75 (line 378) or 100 µM (line 394) for intravitreal injection (in vivo experiment)?
Lines 397-399: State what the significance was?
Line 403: Add a dose of SA-10.
Line 438: Add scale bars.
Line 426: The results of the SOD1 intensity quantifications are missing (in Figure 7), since you stated in the M&M that the results of this are presented as a mean ± SEM.
Author Response
Comments and Suggestions for Authors
The authors conducted a extensive study and determined that the novel hybrid compound SA-10 has a protective effect on retinal neurons by reducing oxidative stress and inflammatory cytokines. I have some minor considerations (see the comments below). After these are addressed, I would recommend the manuscript for publication.
Abstract
Reduce the number of words in the abstract (420 words compared to the 200 allowed).
Response: Thank you for your kind review. The abstract has been reduced to the 200 word limit as allowed.
Line 23: In the main text, the authors did not mention that hybrid molecule SA-10 was tested in concentrations of 1 µM as stated in the abstract.
Response: Corrected
It is not clear why the authors used SA-2 in this study. Please explain.
Response: In this study, we utilized compound SA-10 as our primary test compound and not SA-2. The rationale behind selecting SA-10 was primarily based on previous investigations conducted in our laboratory.
In comparison to the first generation hybrid compound SA-2, compound SA-10 exhibited superior (25%) superoxide scavenging activity in NTM-5 cell supernatants and showed a more potent and robust scavenging activity of hypochlorouos acid radical and peroxynitrite free radical (Amankwa C,E.,et al, 2021, DOI: 10.3390/antiox10040575).
Additionally, another study provided supporting evidence for the effectiveness of compound SA-10. In an acute murine model of ischemia/reperfusion (I/R), SA-10 significantly reduced muscle damage, mitigated hyperinflammation, and reduced lung edema three days after administration. Furthermore, significant improvements in blood perfusion and physical endurance were observed over a period of 30 days (p< 0.05) in the chronic ischemia murine model (Hinkle et al, 2021, Nanomedicine p. 102400).
Materials and Methods
Line 176: State the cytokines and chemokines included in the multiple assay kit.
Response: All information added.
Results
Line 279: Is the 1.5-fold increase significant? If it is not, there is no point in stating it.
Response: Edited as suggested. Thanks
Line 282: “...OGD/R.due…” remove the dot.
Response: We greatly appreciate your critical evaluation. This concern has been fixed. The dot has been removed.
Line 284-285: Authors point that SA-10 significantly increased cell proliferation compared OGD/R+SIN-1. Mark in the Figure 2B. The same applies to the following sentence.
Response: Corrected
Line 289: It is not indicated in Figure 2D that SA-10 enhances proliferation as indicated in the section Results.
Response: Corrected
Line 291: in Figure 2D move “OGD/R” in the level of SIN-1A and SIN-10.
Response: Corrected
Line 293: Please indicate more clearly that in both tests (cytotoxicity and proliferation) 2 types of cells were used.
Response: The use of R28 cell lines and primary rat retinal mixed neurons have been provided in the figure 2 legend.
Lines 300-306: I think the methodology for these results is lacking. Please check.
Response: WE have added all information on the mRNA isolation and qPCR method in the method section
Lines 320-321: I suggest you rather point out that SA-10 significantly reduced the level of ROS.
Response: Thank you for pointing this out. We have provided clarity on the ROS reduction effect of SA-10 as recommended.
Line 326: “..only NO donor SIN-1 produces higher amount of NO (Fig.4B)..”
This is debatable, given that there is no difference between SIN-1 and SIN-10 and both compounds significantly reduced NO production in rat microglia cells.
Response: We deleted this sentence to avoid the ambiguity
Line 360: I suggest removing “ns” in Figure 5a and 5b, (it is understood if there are no asterisks that it is not significant) to make the figure more understandable.
Response: Corrected
Did the authors use a dose of 75 (line 378) or 100 µM (line 394) for intravitreal injection (in vivo experiment)?
Response: It is corrected to 78 µM for SA-10
Lines 397-399: State what the significance was?
Response: Thank you for your kind notice. We have included the groups that we were referring to have been significant for the a- and b- wave amplitude.
Line 403: Add a dose of SA-10.
Response: Thank you. A dose of 2% SA-10 has been added.
Line 438: Add scale bars.
Response: Added
Line 426: The results of the SOD1 intensity quantifications are missing (in Figure 7), since you stated in the M&M that the results of this are presented as a mean ± SEM.
Response: Added the bar graphs and stats to Figure 7.
Reviewer 3 Report
Comments and Suggestions for Authors
In the manuscript: “Neuroprotective and anti-inflammatory activities of hybrid small molecule SA-10 in ischemia/reperfusion induced retinal neuronal injury models,” the authors examine the protective effect of SA-10 in retinal neurons by decreasing oxidative stress, and inflammatory cytokines. It is an interesting topic that contributes to knowledge in the area, but certain issues must be corrected.
1. In the abstract and introduction, the authors must mention the gap the manuscript will fill within the current knowledge.
2. In the abstract, the authors mention that treatment with TII (a cocktail of TNF-22 α, IL-1β, and IFN-Ƴ) for 1-hour mimics I/R injury; why do the authors mention this? TII treatment is a treatment for inducing similar conditions to I/R; please explain why or why not. Note that the reference where this treatment is mentioned talks about the induction of inflammation, not the production of I/R-like conditions.
3. The results must be improved by adding an explanation at the beginning of their description. Also, the authors must mention why the assay in question was used.
4. The results must have concluded. That is, authors must mention the conclusions of each result: for instance, these results together suggest… or we concluded that…
5. The authors must mention all the reagents used during their research in section 2.1 Chemical and Reagents, mentioning the manufacturing company and catalog number.
6. The authors must mention what method they used to induce hypoxia since, in the materials section, they do not mention their method.
Minor revisions
1. Define all abbreviations to be presented in the text. For instance, MTT.
2. Revise grammar.
Comments on the Quality of English Languageno comments
Author Response
Comments and Suggestions for Authors
In the manuscript: “Neuroprotective and anti-inflammatory activities of hybrid small molecule SA-10 in ischemia/reperfusion induced retinal neuronal injury models,” the authors examine the protective effect of SA-10 in retinal neurons by decreasing oxidative stress, and inflammatory cytokines. It is an interesting topic that contributes to knowledge in the area, but certain issues must be corrected.
- In the abstract and introduction, the authors must mention the gap the manuscript will fill within the current knowledge.
Response: The gap in literature was highlighted in the introduction section briefly. Moreover, additional rationale for the identified research gap has been incorporated into the introduction section to underscore the significance and necessity of this study.
“Currently there is no cure for ischemia/reperfusion induced retinal injury other than the standard therapeutic approach of restoring blood flow to salvage the hypo perfused tissues. However, efficient reperfusion of the tissue is not guaranteed and the limited tolerance of neurons to hypoxic stress imposes a restricted time window for effective reperfusion therapy [7, 8]. Moreover, studies examining the consequence of inflammatory gene expression and therapeutic alternatives to retinal IR injury are limited. Hence, there re-mains an unmet need to identify potential therapeutic agents that can address the intricate pathophysiology of ischemia/reperfusion retinal injury
- In the abstract, the authors mention that treatment with TII (a cocktail of TNF- α, IL-1β, and IFN-Ƴ) for 1-hour mimics I/R injury; why do the authors mention this? TII treatment is a treatment for inducing similar conditions to I/R; please explain why or why not. Note that the reference where this treatment is mentioned talks about the induction of inflammation, not the production of I/R-like conditions.
Response: We appreciate your thoroughness and precision in improving the clarity of the content. The rephrasing now accurately conveys that the TII treatment led to a significant inflammatory response, consequently triggering microglial activation.
- The results must be improved by adding an explanation at the beginning of their description. Also, the authors must mention why the assay in question was used.
Response: Thank you. We have incorporated conclusions as suggested.
- The results must have concluded. That is, authors must mention the conclusions of each result: for instance, these results together suggest… or we concluded that…
Response: Thank you. We have incorporated the recommended conclusions.
- The authors must mention all the reagents used during their research in section 2.1 Chemical and Reagents, mentioning the manufacturing company and catalog number.
Response: Thank you. We have incorporated all the reagents as advised in Section 2.1.
- The authors must mention what method they used to induce hypoxia since, in the materials section, they do not mention their method.
Response:
The cellular hypoxia (ischemia) was described in Oxygen glucose deprivation (OGD)/reperfusion methods in section 2.5 and in vivo mouse model of I/R injury in section 2.10
Minor revisions
- Define all abbreviations to be presented in the text. For instance, MTT.
- Revise grammar.
Response:
Thank you for your thoughtful revision. All abbreviations have been appropriately expanded as presented in the text, and grammar has been meticulously corrected.
Round 2
Reviewer 3 Report
Comments and Suggestions for Authors
The authors have responded satisfactorily to my reviews. Thank you!
Comments on the Quality of English LanguageNo comments
Author Response
Thank you!